# Predictive State Propensity Subclassification (PSPS): A causal inference algorithm for data-driven propensity score stratification

**Joseph Kelly**                                     JOSEPHKELLY@GOOGLE.COM
*Google, New York, NY 10023, USA*

**Jing Kong**                                          JINGKONG@GOOGLE.COM
*Google, Mountain View, CA 94043, USA*

**Georg M. Goerg**[*]                                  GEORG@EVOLUTIONIQ.COM
*EvolutionIQ, New York, NY 10010, USA*

## Abstract

We introduce Predictive State Propensity Subclassification (PSPS), a novel learning algorithm for causal inference from observational data. PSPS combines propensity and outcome models into one encompassing probabilistic framework, which can be jointly estimated using maximum likelihood or Bayesian inference. The methodology applies to both discrete and continuous treatments and can estimate unit-level and population-level average treatment effects. We describe the neural network architecture and its TensorFlow implementation for likelihood optimization. Finally we demonstrate via large-scale simulations that PSPS outperforms state-of-the-art algorithms – both on bias for average treatment effects (ATEs) and RMSE for unit-level treatment effects (UTEs).

**Keywords:** causal inference; multi-task learning; predictive state representation; propensity score matching; causal representation learning.

## 1. Introduction

Measuring the impact of an intervention on an outcome of interest is ubiquitous across science, business, and engineering. For example, researchers want to find out whether a new drug has the desired effect on blood pressure reduction; advertisers would like to know whether an ad worked to increase sales; or a software company wants to evaluate the impact of a new feature launch on improved user experience. In an ideal world a randomized experiment would be run to estimate the impact of each of these interventions. Randomization ensures balance in expectation on important confounders – variables that affect both who got treated and the outcome – across the different treatment arms allowing for unbiased estimates of the causal effect of the intervention. An experimenter may furthermore block or stratify on certain features in order to ensure balance across treatment assignments for a particular randomization (Bernstein, 1927).

Randomized experiments, however, are not always feasible due to time, cost, legal, or ethical reasons – to name a few – in which case the decision maker has to rely on observational data. Their lack of randomization leads to feature imbalances between the treatment arms and thus make identification and estimation of causal effects difficult. Causal inference algorithms for observational data cannot rely on randomization, but have to use other means to mitigate the imbalances. As such they are often viewed as tools to help fix broken randomized experiments (Rubin, 2008), by trying to re-create balance between different treatment levels that did not occur due to the lack of randomization.

---

[*] Majority of Georg M. Goerg's contributions to this work were completed at Google.

It is important to emphasize that in order to make good data-driven decisions on how to intervene in real world settings, algorithms need to provide unbiased and low-variance causal effect estimates. Learning algorithms that solely optimize for predictive accuracy are likely diametrical to accurate causal effect estimates as they fall victim to selection bias & confounding (Bareinboim et al., 2014).

In this work we introduce *Predictive State Propensity Subclassification* (PSPS), a novel causal learning algorithm that leverages multi-task learning (Ruder, 2017). PSPS follows the Rubin Causal Model (RCM) framework (Imbens and Rubin, 2010) and is in the same vein as matching, IPW and traditional subclassification that aim to fix the broken design of the experiment. Our key contributions are three-fold: a) we incorporate predictive state smoothing (PRESS) (Goerg, 2017, 2018) for the propensity score, which yields a principled data-driven way to obtain the *strata* or *blocks* used in traditional matching approaches; b) the entire causal inference algorithm can be represented as an end-to-end multi-task neural network, which allows for – statistically and computationally – efficient training and inference through a single computational graph; c) we derive a causal loss function that balances predictive performance with properly accounting for selection bias.

## 1.1. Literature Review

Existing methods that aim to fix non-randomization in observational data are often based on matching (Stuart, 2010) or inverse propensity weighting (IPW) coupled with outcome modeling; e.g., entropy balancing (Hainmueller, 2012) and double robust estimation (Robins and Rotnitzky, 1995; Hirano et al., 2003; Lunceford and Davidian, 2004). *Subclassification* addresses feature imbalance by creating strata of observations such that within each stratum there is approximate feature balance between treated and control units (Cochran, 1968). We do not use *do*-calculus (Pearl, 1995) in this work, but since PSPS is a fully probabilistic framework it is straightforward to use plug-in causal predictions and conditional and joint distribution estimates to a *do*-calculus based causal analysis.

Recently causal inference has also received attention in the machine learning community: for example via counterfactual representation learning (Johansson et al., 2016) or *Dragonnet* (Shi et al., 2019). Section 4 compares PSPS to various methods on causal benchmark datasets.

## 2. Methodology, Neural Network Architecture & Learning Objectives

Let $T$ be the treatment variable, $Y$ the observed outcome, and $\mathbf{X}$ is a collection of pre-treatment features. For example, the canonical causal estimation scenario is the case of binary treatment, $T \in \{0,1\}$, with continuous outcome $Y \in \mathbb{R}$, and $k$-dimensional feature space $\mathbf{X} \in \mathbb{R}^k$. We will revisit an example of continuous treatment in Sections 3. Unless specified otherwise though, we intentionally leave the dimensionality and variable types of $(T, Y, \mathbf{X})$ unspecified as the core concepts and derivations also hold for multivariate outcomes and multi-level or continuous treatments.

### 2.1. Factorization of the Joint Distribution of Treatment & Outcome

Statistically, treatment assignment and application are confounded when learning from non-randomized data. In essence many causal methods aim to break this confounding between assignment and the application effect of treatment on the outcome. In PSPS this dual role is made explicit by

$$p(Y,T \mid \mathbf{X}) = p_{\theta_Y}(Y \mid T,\mathbf{X}) \cdot p_{\theta_T}(T \mid \mathbf{X}) = p_{\theta_Y}(Y \mid T_{apply},\mathbf{X}) \cdot p_{\theta_T}(T_{assign} \mid \mathbf{X}), \tag{1}$$

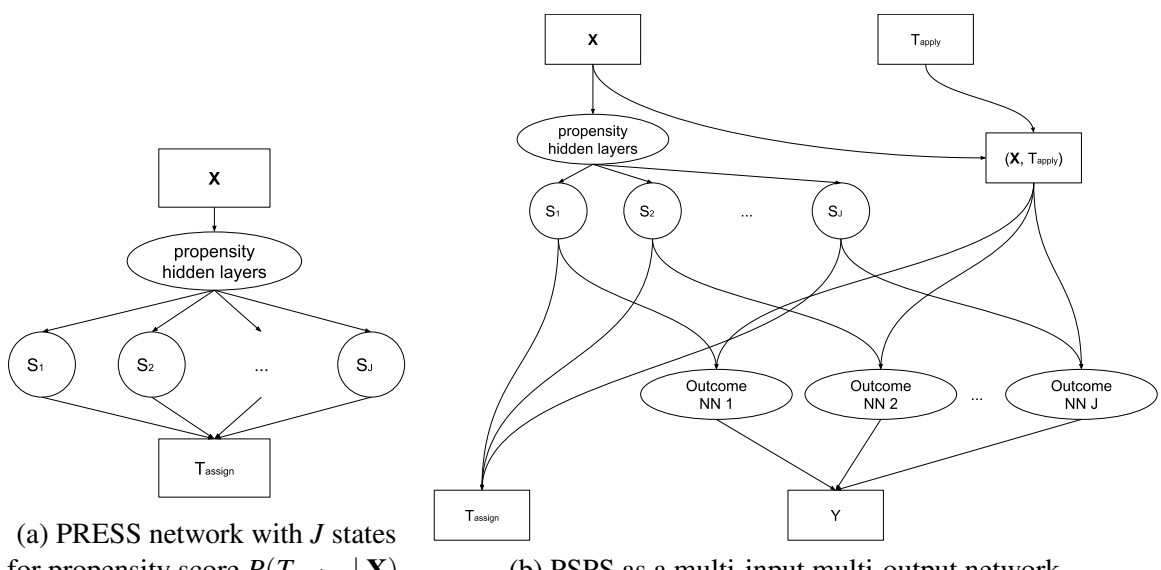

(a) PRESS network with *J* states for propensity score $P(T_{assign} \mid \mathbf{X})$.

(b) PSPS as a multi-input multi-output network.

Figure 1: PSPS model architecture: propensity model $P(T \mid \mathbf{X})$ with *J* states and the full joint model $P(Y, T \mid \mathbf{X})$, where $\mathbf{X}$ are features, *T* is the treatment, and *Y* is the outcome; each state $s_j$ has a separate outcome model for $P(Y \mid \mathbf{X}, S = s_j)$. The dual role of treatment *T* in (1) is explicit as 2 distinct nodes $T_{apply}$ & $T_{assign}$ in the computational graph.

where $\theta_Y$ and $\theta_T$ parameterize the conditional distributions of *Y* and *T*, respectively. The factorization in (1) makes it explicit that *T* plays two roles: i) once as the output of a propensity model (treatment assignment, $T_{assign}$), and ii) as an input to an outcome model (treatment application, $T_{apply}$). Since assignment and application are confounded we modulate the distribution on the observed outcome ($Y \mid T_{apply}, \mathbf{X}$) with the fact that there was self-selection to treatment ($T_{assign} \mid \mathbf{X}$). This distinction also illustrates that practicioners are truly interested in the effect of *applying* treatment *t* on unit *i* with observed features $\mathbf{x}_i$, i.e. $p_{\theta_Y}(Y_i \mid T_{apply} = t, \mathbf{X} = \mathbf{x}_i)$. While treatment assignment and application are the same random variable and data, respectively, we keep the distinction for clarity on the computational graph. Figure 1 shows that $T_{assign}$ and $T_{apply}$ play two different roles: $T_{assign}$ is an output while $T_{apply}$ is an input. It is important to have separate nodes – rather than a single "*T*" node – as this would result in an incorrect computational representation, and hence implementation.

### 2.2. Propensity Score Subclassification

Under a randomized-block experiment the propensity score $P(T \mid \mathbf{X})$ is known by construction and the properties of randomization meet the ignorable treatment assignment ("ignorability") assumption (Rosenbaum and Rubin, 1983a) allowing for the identification and estimation of causal effects.[1] For observational data studies treatment assignment is typically unknown and as such the ignorability assumption is usually made on a leap of faith after adjusting for any observed imbalances.

---

1. Causal effect identification and the assumptions needed to link potential outcomes under different treatments to the observed data are well established and will not be examined in detail (see Rubin, 2005; Imai and van Dyk, 2004, for binary and continuous treatment assumptions, respectively). Although we too make these assumptions, our focus is on creating an estimator with good performance that adjusts for the observed imbalances.

To deal with potential self-selection bias and imbalances between treated and control units, Rosenbaum and Rubin (1985) proposed *subclassification*, which stratifies units by a coarsened value of predicted propensity scores $\hat{P}(T \mid \mathbf{X})$, – usually achieved by binning; it is also often accompanied by a model-based adjustment to deal with any remaining imbalances. The idea being that observations within each bin have approximately the same propensity score and thus are more robust to any misspecification in the outcome models of each bin. The (predicted) outcome for treated and controls can then be compared within each stratum to estimate the effect of treatment.

One common problem with propensity score subclassification is that there are little to no guidelines on how to actually create the subclasses. Common heuristics are to a) bin on quantiles, b) have equally sized bins, or c) try to at least make the largest bin size moderately small. Imbens and Rubin (2015) create bins recursively by testing equality of propensity score means between treated and control via a t-test. If the difference is significant then the stratum is split at the median and the procedure repeats on sub-strata until the test statistic is not significant or a pre-specified minimum number of treated and controls are met. The end result, however, is inevitably an ad-hoc heuristic.

### 2.3. Predictive States As The Optimal Data-Driven Subclassification

PSPS follows the same experimental design as traditional subclassification except for how subclasses are created and how subsequent analyses take place. The core causal assumptions such as "ignorability" are the same and we rely on these established results (Imbens and Rubin, 2015).

PSPS puts the goal of constructing subclasses front and center by learning the optimal stratification directly from the observed data using Predictive State Smoothing (PRESS) (Goerg, 2017, 2018). PRESS represents the conditional predictive distribution $p(T \mid \mathbf{X})$ as a mixture over $J$ latent predictive states, $\{s_1(\mathbf{X}), \ldots, s_J(\mathbf{X})\}$ (Figure 1). Each state is an equivalence class over distribution space such that observations with the same conditional distribution are in the same predictive state. That is $\mathbf{X}_{i_1}$ and $\mathbf{X}_{i_2}$ have the same state $s_j$ if and only if

$$p(T \mid \mathbf{X}_{i_1}) \equiv p(T \mid \mathbf{X}_{i_2}) = p(T \mid s_j). \tag{2}$$

The key insight here is that predictive states are exactly the strata $\{s_1, \ldots, s_J\}$ that would have ideally been blocked on when assigning treatment: knowing the predictive state of observation $i$ – which is unobserved in observational data –would be sufficient to knowing the propensity score distribution of all observations in that state. Predictive states are also minimal sufficient for prediction, i.e., they make $T$ conditionally independent of $\mathbf{X}$ given $s_j$ (Goerg, 2017). Hence

$$p(T \mid \mathbf{X}) = \sum_{j=1}^{J} p(S = s_j \mid \mathbf{X}) \cdot p(T \mid S = s_j, \mathbf{X}) = \sum_{j=1}^{J} w_j(\mathbf{X}) \cdot p(T \mid S = s_j), \tag{3}$$

where $w_j(\mathbf{X}) := p(S = s_j \mid \mathbf{X})$ denote the probability that $\mathbf{X}$ belongs to state $s_j$.

We use the weight-vector notation $\mathbf{w} = (w_1, \ldots, w_J)$ as the representation of the state space mapping; by definition $\sum_{j=1}^{J} w_j = 1$ and $w_j \geq 0$. It is easy to see that two observations $i_1$ and $i_2$ – with potentially different features $\mathbf{X}_{i_1}$ and $\mathbf{X}_{i_2}$ – that are mapped to the same predictive state with same probabilities, i.e., $\mathbf{w}_{i_1} \equiv \mathbf{w}_{i_2}$, necessarily have the same treatment distribution since $p(T_i \mid \mathbf{X}_i) = \sum_{j=1}^{J} w_{i,j} \cdot p(T_i \mid s_j)$. Put in other words, if $i_1$ and $i_2$ have different observed features, $\mathbf{x}_{i_1}$ and $\mathbf{x}_{i_2}$, but are in the same state, $s_j$, then they have the same propensity score:

$$P(T_i \mid \mathbf{X} = \mathbf{x}_{i_1}, S = s_j) = P(T_i \mid S_j) = P(T_j \mid \mathbf{X} = \mathbf{x}_{i_2}, S = s_j). \tag{4}$$

This is exactly the underlying goal of traditional subclassification by Imbens and Rubin (2015). Another way of thinking about it is that each state is a matched group where matching occurs based on an optimal distance between units $i_1$ and $i_2$ with features $\mathbf{X}_{i_1}$ and $\mathbf{X}_{i_2}$ measured as the distance between propensity score distributions $P(T_{i_1} \mid \mathbf{X}_{i_1})$ and $P(T_{i_2} \mid \mathbf{X}_{i_2})$.

## 2.4. Combining Outcome Models Across Predictive States

After constructing the subclasses, PSPS follows the traditional subclassifcation philosophy of training an outcome model in each subclass (Figure 1). This has the effect of adjusting for features if the PRESS propensity model is misspecified. Even when it is correctly specified, more accurate outcome models can further reduce the variability in causal effect estimates (Rosenbaum, 2002).

To replicate the behavior of estimating the causal effect within each subclass we use the propensity model predictive states to factorize the conditional outcome distribution in (1) as

$$
\begin{aligned}
p(Y_i \mid T_i, \mathbf{X}_i) &= \sum_{j=1}^{J} p(S = s_j \mid \mathbf{X}_i, T_i) \times p(Y_i \mid T_i, \mathbf{X}_i, S = s_j) \\
&= \sum_{j=1}^{J} \frac{p(T_i \mid S = s_j)}{p(T_i \mid \mathbf{X}_i)} \cdot P(S = s_j \mid \mathbf{X}_i) \times p_{\theta_Y^{(j)}}(Y_i \mid T_i, \mathbf{X}_i), \\
&= \sum_{j=1}^{J} (\rho_{i,j} \cdot w_{i,j}) \times p_{\theta_Y^{(j)}}(Y_i \mid T_i, \mathbf{X}_i),
\end{aligned}
\tag{5}
$$

where the lift ratio $\rho_{i,j} \in [0, \infty)$ quantifies how much the state predictive distribution, $p(T_i \mid s_j)$, differs from the point prediction of the propensity score for feature $\mathbf{x}_i$, $p(T_i \mid \mathbf{X}_i = \mathbf{x}_i)$, and $\theta_Y^{(j)}$ parameterizes the outcome model in state $j$. Since $v_{i,j} := \rho_{i,j} \cdot w_{i,j} = P(S = s_j \mid \mathbf{X}_i, T_i)$ is a proper probability mass function for each $i$, the decomposition in (5) is a mixture over state-conditional outcome models, $p_{\theta_Y^{(j)}}(Y \mid T, \mathbf{X})$, where model $j$ contributes with weight $v_{i,j}$ to the final estimate for unit $i$. That means observation $i$ is only partially contributing to state $j$ and also explains why, in general, outcome model parameter estimates will be different for every state.[2]

Consider the special case where each unit $i$ is mapped exactly to one state with probability 1 (i.e., a deterministic mapping), then also $\rho_{i,j} = 1$ for exactly one $j$ and 0 otherwise. In that case,

$$
p(Y_i \mid T_i, \mathbf{X}_i) = \sum_{j=1}^{J} p_{\theta_Y^{(j)}}(Y_i \mid T_i, \mathbf{X}_i, S = s_j) \cdot \mathbb{1}(S = s_j)_{\mathbf{X}_i},
\tag{6}
$$

which is equivalent to binning propensity scores and training an outcome model in each bin. Thus, PSPS is a probabilistic generalization of traditional – and often heuristic – propensity score binning.

## 2.5. Model Training via Multi-Task Learning

Since PSPS provides a consistent causal probabilistic framework, we use maximum likelihood estimation (MLE) as the primary learning strategy; we leave Bayesian inference for future work.

---

2. Each state could be equipped with their own model class, e.g., a Random Forest in $j_1$, a linear model in $j_2$, and a deep net in $j_3$. In practice, they are usually of the same model family – with different (estimated) parameters per state.

Decomposing the joint distribution in (1) allows us to specify propensity model and outcome models separately, yet link them through their joint log-likelihood function as

$$\ell(\boldsymbol{\theta}_Y, \boldsymbol{\theta}_T; Y, \mathbf{X}, T) = \ell(\boldsymbol{\theta}_Y; Y, \mathbf{X}, T_{apply}) + \ell(\boldsymbol{\theta}_T; \mathbf{X}, T_{assign}), \tag{7}$$

where $\ell(\tau; Z) := \log p_\tau(Z)$ is the log-likelihood of $\tau$ for data $Z$. This decomposition, also used for Targeted MLE (TLME) (Van der Laan and Rose, 2011), naturally leads to framing parameter estimation as a multi-task learning problem (Ruder, 2017) with a negative log-likelihood loss

$$L(Y, T, \hat{Y}(\boldsymbol{\theta}_Y, \boldsymbol{\theta}_T; \mathbf{X}, T_{apply}), \hat{T}_{assign}(\boldsymbol{\theta}_T; \mathbf{X})) = L(Y, \hat{Y}(\boldsymbol{\theta}_Y, \boldsymbol{\theta}_T; \mathbf{X}, T_{apply})) + L(T_{assign}, \hat{T}_{assign}(\boldsymbol{\theta}_T; \mathbf{X})),$$
$$\text{joint loss} = \text{outcome model loss} + \text{propensity model loss}, \tag{8}$$

where $\hat{T}_{assign}(\boldsymbol{\theta}_T; \mathbf{X})$ is the predicted propensity of assigning treatment and $\hat{Y}(\boldsymbol{\theta}_Y, \boldsymbol{\theta}_T; \mathbf{X}, T_{apply})$ is the predicted outcome when applying treatment. The MLE can be obtained by solving

$$\widehat{(\boldsymbol{\theta}_Y, \boldsymbol{\theta}_T)}_{MLE} = \arg \min_{\boldsymbol{\theta}_Y, \boldsymbol{\theta}_T} L(Y, T, \hat{Y}(\boldsymbol{\theta}_Y, \boldsymbol{\theta}_T; \mathbf{X}, T_{apply}), \hat{T}_{assign}(\boldsymbol{\theta}_T; \mathbf{X})). \tag{9}$$

For randomized experiments the second term (8) is constant since $T$ is independent $\mathbf{X}$; hence PSPS reduces to a simple outcome model when faced with properly randomized treatment.

### 2.6. Separation of Design and Analysis

In observational studies, practitioners often face a choice between only training an outcome model ($P(Y \mid T, \mathbf{X})$) or a two-step procedure of training a propensity model first ($P(T \mid \mathbf{X})$) and an adjusted/weighted outcome model later. PSPS includes both approaches as special cases, and allows practitioners to gradually move between them using a tunable hyperparameter.

The decomposition in (8) can be seen as trying to find an optimal outcome model, but with a penalty for bad propensity models. It is then natural to introduce a penalty parameter, $\lambda \in [0, \infty)$,

$$L(Y, \hat{Y}(\boldsymbol{\theta}_Y, \boldsymbol{\theta}_T)) + \lambda \cdot L(T, \hat{T}(\boldsymbol{\theta}_T)), \tag{10}$$

which reduces to (8) when $\lambda = 1$. This can be interpreted as just training a single outcome model $Y \mid T, \mathbf{X}$, with a penalty $L(T, \hat{T}(\boldsymbol{\theta}_T))$ that adjusts for feature distribution imbalances between treated and controls (see also counterfactual representation learning in Johansson et al., 2016).

When $\lambda = 0$ the second term vanishes from (10) and the optimizer only focuses on outcome models, weighted by a random mixture of predictive state weights from each state, which – in expectation – is just a simple overall outcome model. For $\lambda \to \infty$ the optimizer first tries to obtain optimal propensity score, and only then will it further tune the outcome model parameters to improve total loss (at the given optimal propensity score loss). Thus for $\lambda \to \infty$ optimizing (10) approximates the standard two-step approach that separates design and analysis stages of an experiment.

### 2.7. Binary Treatment & Gaussian Outcomes

In this section we describe parameter estimation for binary treatment and a continuous, univariate outcome.[3] It is also this scenario that we evaluate through simulation studies in Section 4.

---

3. Other scenarios such as continuous treatment can be handled analogously by adapting the likelihood functions.

**Algorithm 1:** Two-step iterative training for PSPS
**Input:** number of states $J$; data: treatment $\mathbf{t}$; features $\mathbf{X}$; outcome $\mathbf{y}$
**Output:** trained models: 1 propensity model and $J$ outcome models

1. Train PRESS model for $P(T \sim \mathbf{X})$ with $J$ predictive states $\{s_1, \ldots s_J\}$.

2. Compute predicted weights $w_{i,j} = \hat{p}(s_j \mid \mathbf{X}_i)$ and $v_{i,j} = \hat{p}(s_j \mid \mathbf{X}_i, T_i)$.

3. for $j = 1, \ldots, J$: use weights $v_{i,j}$ to train a *weighted* outcome model for $P(Y \mid \mathbf{X}, T)$.

   The outcome models could be a neural net, a regression, or even just the sample mean of the response; $Y \mid \mathbf{X}, T = 1$ and $Y \mid \mathbf{X}, T = 0$ can also be modeled separately.

---

For binary treatment the negative log-likelihood for observation $i$ is binary cross-entropy

$$L(T_i, \hat{T}_i(\boldsymbol{\theta}_T)) = - \left[ T_i \cdot \log(\hat{T}_i(\boldsymbol{\theta}_T)) + (1 - T_i) \cdot \log(1 - \hat{T}_i(\boldsymbol{\theta}_T)) \right]. \tag{11}$$

For univariate continuous outcomes a Normal log-likelihood, $N(y \mid x, \sigma_\varepsilon^2)$, is a natural candidate leading to a weighted sum of outcome losses for each state:

$$L(Y_i, \hat{Y}_i(\boldsymbol{\theta}_Y); \boldsymbol{\theta}_T, \mathbf{X}_i) = - \sum_{j=1}^{J} \log(\sigma_{\varepsilon,j}) + \sum_{j=1}^{J} v_{i,j}(\mathbf{X}_i, \boldsymbol{\theta}_T) \cdot \frac{(Y_i - \hat{Y}_i(\boldsymbol{\theta}_{Y,j}))^2}{2\sigma_{\varepsilon,j}^2}, \tag{12}$$

where $v_{i,j}(\mathbf{X}_i, \boldsymbol{\theta}_T)$ is the probability of unit $i$ belonging to state $j$, $\hat{Y}_i(\boldsymbol{\theta}_Y)$ is the outcome model prediction for $Y_i$, and $\boldsymbol{\theta}_Y = \{(\boldsymbol{\theta}_j, \sigma_{\varepsilon,j}) \mid j = 1, \ldots, J\}$ are all free parameters of the outcome model, with $\sigma_{\varepsilon,j} \in \boldsymbol{\theta}_Y$ as the unknown standard deviation of the $j$-th outcome model residual, $\varepsilon_j := Y_i - \hat{Y}_i(\boldsymbol{\theta}_{Y,j})$. The average loss across $N$ observations equals

$$\frac{1}{N} \sum_{i=1}^{N} L(Y_i, \hat{Y}_i(\boldsymbol{\theta}_Y); \boldsymbol{\theta}_T, \mathbf{X}_i) - \frac{1}{N} \sum_{i=1}^{N} \left[ T_i \cdot \log(\hat{T}_i(\boldsymbol{\theta}_T; \mathbf{X}_i)) + (1 - T_i) \cdot \log(1 - \hat{T}_i(\boldsymbol{\theta}_T; \mathbf{X}_i)) \right]. \tag{13}$$

It is worth to point out that the PRESS propensity model parameters, $\boldsymbol{\theta}_T$, do not only enter (13) via the binary crossentropy propensity loss in (11), but also affect the outcome loss (12) through the weighting over the $J$ states, $v_{i,j}(\mathbf{X}_i, \boldsymbol{\theta}_T)$.

Algorithm 1 describes an iterative learning procedure to optimize (13). However, since PSPS models the joint distribution directly, it can be expressed as a single computational graph using standard neural network architectures (Fig. 2.1). This has the benefit of training all parameters and predictive states jointly, which avoids the – statistically and computationally – inefficient iterative procedure. In particular, the proposed MLE optimization and causal effect inference runs entirely within a single TensorFlow graph (Abadi et al., 2016), which in turn can be easily incorporated into larger graphs that rely on causal inference as part of their computation.

## 3. Estimating Causal Effects

Once the predictive state mapping and all outcome models have been trained, unit-level treatment effect (UTE) can be obtained as a weighted average of UTEs within each state:

$$\hat{\Delta}_i^{t+\varepsilon} = \sum_{j=1}^{J} \hat{v}_{i,j} \cdot \Delta_i^{t+\varepsilon}(j) = \sum_{j=1}^{J} P(s_j \mid \mathbf{x}_i, \hat{\theta}_T) \cdot \left( \mathbb{E}(y \mid t_i = t+\varepsilon, \mathbf{x}_i; \hat{\theta}_j) - \mathbb{E}(y \mid t_i = t, \mathbf{x}_i; \hat{\theta}_j) \right), \quad (14)$$

where $\hat{\Delta}_i(j)$ is the estimated UTE for unit $i$ from outcome model $j$, and $\varepsilon$ is discrete (continuous) for categorical (continuous) treatment. Estimates for sample, population and conditional effects can be obtained by aggregating over the appropriate (sub-)sample of observations. For example, average treatment effect (ATE) and average treatment effect on the treated (ATT) can be estimated as

$$\hat{\Delta}_{\text{ATE}}^{t+\varepsilon} = \frac{1}{N} \sum_{i=1}^{N} \hat{\Delta}_i^{t+\varepsilon} \quad \text{and} \quad \hat{\Delta}_{\text{ATT}}^{t+\varepsilon} = \frac{1}{\sum_{i=1}^{N} T_i} \sum_{i=1}^{N} T_i \cdot \hat{\Delta}_i^{t+\varepsilon}. \quad (15)$$

### 3.1. Uncertainty Estimates

While in predictive learning tasks point predictions are of primary interest, in causal inference tasks getting accurate uncertainty estimates around the causal effect are equally – if not more – important. A good causal estimator produces narrow confidence intervals with proper coverage. For simulations in Section 4, we use standard bootstrap procedures (Efron and Tibshirani, 1986) to obtain empirical confidence intervals for $\Delta_i$ and $\Delta_{ATE}$ in (14) and (15), respectively.

## 4. Model Evaluation: Simulation Study

We compare PSPS to several state-of-the-art causal inference methods: an elastic net linear outcome model (Zou and Hastie, 2005), doubly robust inverse propensity weighted (DRIPW) models (Chan et al., 2010) – both linear and Random Forest (Breiman, 2001) versions –, entropy balancing (Hainmueller, 2012), a PRESS outcome model (Goerg, 2017)[4] and *Dragonnet* (Shi et al., 2019).[5]

All of the above algorithms – with the exception of entropy balancing – can not only estimate the average treatment effect (ATE), but also unit-level treatment effects (UTE) (this is equivalent to the conditional average treatment effect (CATE) in our setup). We conduct a simulation study on standard causal inference benchmarks and examine performance on ATE bias and UTE RMSE:

$$\text{ATE bias} = \Delta_{ATE} - \hat{\Delta}_{ATE} \text{ and RMSE UTE} = \sqrt{\frac{1}{N} \sum_{n=1}^{N} (\Delta_i^{0 \to 1} - \hat{\Delta}_i^{0 \to 1})^2}. \quad (16)$$

For each simulation we train a $J = 10$ PSPS model on $n = 1,000$ samples with an 80/20 train/test split and early stopping. Figures 2 and 3 depict each run from a single simulated dataset as a point in the graph. The boxplots give a sense of bias and spread of ATE, where a method with the same

---

4. This PRESS outcome model is not tied to the PRESS architecture underlying the PSPS propensity model. It is simply an alternative neural network outcome model for $P(Y \mid \mathbf{Z})$, with $\mathbf{Z} = (\mathbf{X}, T)$ as the features, which happens to use a PRESS architecture – with entirely different hyperparameters and output activations. The states from this outcome-only model are not related to the PSPS predictive states.

5. To make Dragonnet comparable to PSPS we ensured that both network architectures contain the roughly same number of trainable parameters and only used Scaled Exponential Linear Unit (SELU) activations (Klambauer et al., 2017).

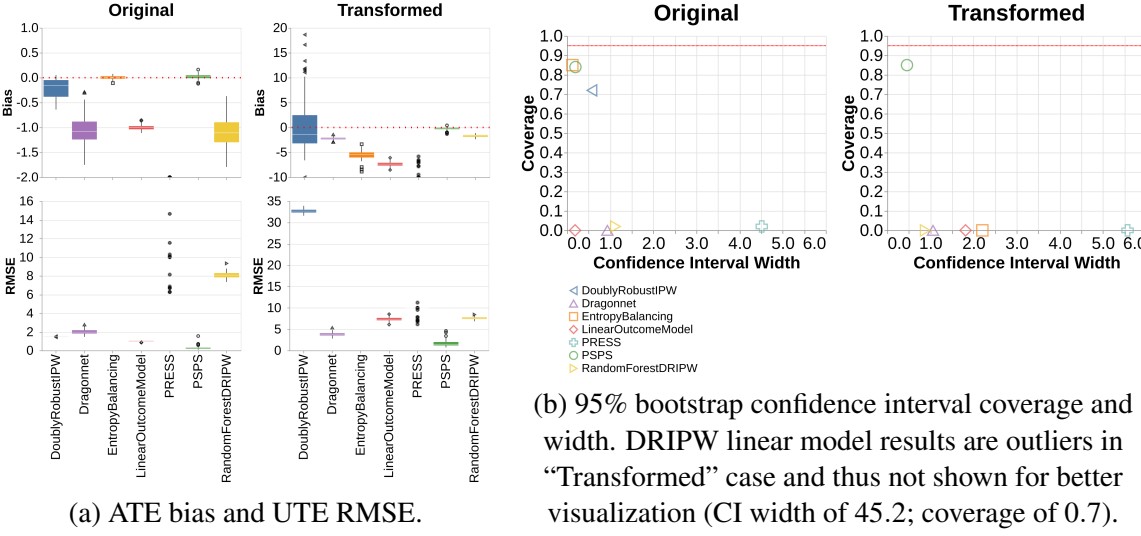

(a) ATE bias and UTE RMSE.

(b) 95% bootstrap confidence interval coverage and width. DRIPW linear model results are outliers in "Transformed" case and thus not shown for better visualization (CI width of 45.2; coverage of 0.7).

Figure 2: Model comparison for Kang & Schafer (Section 4.1)

average (median) bias but smaller variance is preferred. Similarly, the boxplots of RMSE summarize UTE estimation performance for each simulation. This visualization is in lieu of the average bias as by construction the sample mean of the estimated UTEs for each dataset in the simulation is the estimated ATE for that dataset (see (15)); hence the average bias is equivalent to the bias for ATE.

### 4.1. Kang & Schafer: Learning Non-Linear Causal Effects

Kang and Schafer (2007) present a simulation based on a real case study, where four features are drawn independently from a standard Normal, $\mathbf{z} = (z_1, z_2, z_3, z_4) \sim N(0, I_4)$, and the outcome and true propensity score $\pi$ are generated as

$$y = 210 + 27.4z_1 + 13.7z_2 + 13.7z_3 + 13.7z_4 + \varepsilon, \quad \pi = \text{expit}(-z_1 + 0.5z_2 - 0.25z_3 - 0.1z_4)$$

respectively ($\varepsilon \sim N(0, 1)$). The original features $(z_1, z_2, z_3, z_4)$ are then non-linearly transformed as $x_1 = \exp(z_1/2)$, $x_2 = x_2/(1 + \exp(z_1)) + 10$, $x_3 = (z_1 z_3/25 + 0.6)^3$, and $x_4 = (z_2 + z_4 + 20)^2$. This allows us to evaluate causal methods against correctly and incorrectly specified models by either using the original ($\mathbf{z}$) or transformed features ($\mathbf{x}$). This is particularly important for practitioners to see how well methods such as PSPS, PRESS, Dragonnet, and DRIPWRandomForest can approximate – in practice unknown – non-linear dependencies in real world data.

Figure 2 summarizes the results on bias & RMSE. In the "Original" (linear) case, only entropy balancing and PSPS are unbiased; all other methods show medium to strong bias. Interestingly, the elastic net LinearOutcomeModel is slightly biased on the causal estimate with very small RMSE, even though the true response surface is linear in the features. We hypothesize that this can be attributed to incorrectly tuned $\ell_1$ and $\ell_2$ penalties in the elastic net cross validation, as it optimizes for predictive power, not accurate causal estimation.

In the "Transformed" case entropy balancing does significantly worse on ATE bias.[6] PSPS on the contrary is effectively unbiased and has the smallest UTE RMSE amongst all methods. Given

---

6. Recall that entropy balancing is not able to provide UTE estimates.

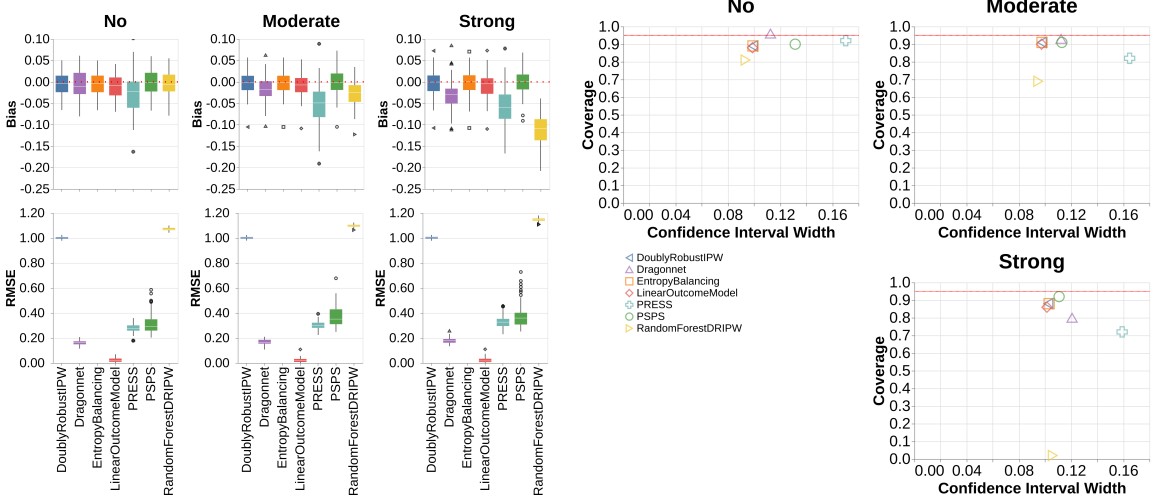

(a) ATE bias (top) and UTE RMSE (bottom)    (b) 95% confidence interval coverage and width

Figure 3: Model comparison for Lunceford & Davidian (Section 4.2)

that in practice, cause and effect are rarely – if ever – linearly related, this comes to show that PSPS is a great choice for estimating causal effects from data collected in real world – non-linear – settings. Figure 2 shows that only PSPS and entropy balancing have close to proper confidence interval coverage with small widths in the "Original" case; in the "Transformed" case PSPS is the only algorithm that has proper confidence intervals. Interestingly, PSPS not only outperforms parametric & linear models but also significantly improves on Dragonnet, which shows significant bias for ATE and is unable to produce confidence intervals at any reliability even in the linear case.

### 4.2. Lunceford & Davidian: Feature Selection

Lunceford and Davidian (2004) proposed another benchmark simulation where the data also consists of $\{(\mathbf{X}_i, \mathbf{Z}_i, T_i, Y_i), i = 1, \cdots, n\}$, but all of them are observed. Both $\mathbf{X}_i$ and $\mathbf{Z}_i$ are three dimensional vectors (see B.1 for details on data generating process for $\mathbf{X}$ and $\mathbf{Z}$). The propensity score is only related to $\mathbf{X}$ – not $\mathbf{Z}$ – through $\pi = \text{expit}\left(\beta_0 + \sum_{j=1}^{d} \beta_j \mathbf{X}_{i,j}\right)$, where $\beta$ controls the association between $\mathbf{X}$ and $T$. We follow the original work and use $\beta^{no} = (0,0,0,0)^T$, $\beta^{moderate} = (0, 0.3, -0.3, 0.3)^T$, and $\beta^{strong} = (0, 0.6, -0.6, 0.6)^T$. The response $Y$ follows

$$Y_i = v_0 + \sum_{j=1}^{3} v_j \mathbf{X}_{ij} + v_4 T_i + \sum_{j=1}^{3} \xi_j Z_{ij} + \varepsilon_i, \quad \varepsilon_i \sim N(0,1), \quad v = (0, -1, 1, -1, 2)^T \quad (17)$$

Similarly to $\beta$, $\xi$ controls the association between $\mathbf{Z}$ and $Y$, $\xi^{no} = (0,0,0)^T$, $\xi^{moderate} = (-0.5, 0.5, 0.5)^T$, and $\xi^{strong} = (-1, 1, 1)^T$. The data generation process yields a true population ATE of $\Delta_{ATE} \equiv 2$. Estimating the ATE accurately is challenging in this simulation scenario given various inter- and intra-dependence of $\mathbf{X}$ and $\mathbf{Z}$, while $\mathbf{Z}$ is only influencing the outcome – not the propensity score.

Figure 3 shows that PSPS, linear DRIPW, entropy balancing, and DRIPW perform equally well in all cases. As for Kang & Schafer this is quite compelling for PSPS as it is on par with parametric linear models, while being able to learn non-linearities if data supports it. Random Forest outcome

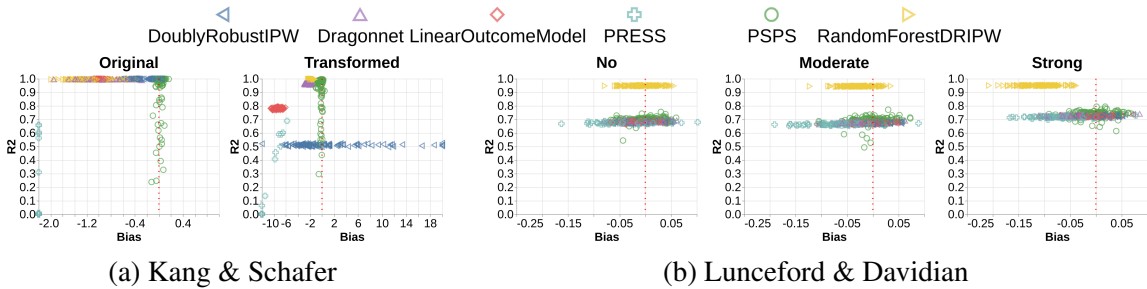

(a) Kang & Schafer  (b) Lunceford & Davidian

Figure 4: $R^2$ of outcome model vs. ATE bias. For marginal bias see Figure 2 and 3.

models and Dragonnet increase in ATE bias for stronger associations. Interestingly, both a linear and Random Forest DRIPW model have difficulty estimating UTEs – compared to competing models with low RMSE. Whilst consistently having the second lowest RMSE (after LinearOutcomeModel) the ATE bias of Dragonnet increases as the association between features and treatment increases. Appendix B.2 has additional results from another causal simulation benchmark (Radcliffe, 2007).

### 4.3. Balancing Outcome and Treatment Model Performance

A key aspect of PSPS is the joint optimization of the (treatment, outcome) likelihood. The decomposition in (10) makes it clear that solely maximizing predictive power for the outcome will produce substantially worse causal estimates compared to an algorithm that strikes a balance between a good treatment model at the cost of some reduction in predictive power in the outcome models.

The simulation study allows us to replicate this empirically. Figure 4 shows the $R^2$ of the underlying outcome model predictions $\hat{y}$ versus the ATE bias for each dataset. As expected, methods with a higher $R^2$ are not guaranteed to achieve a better ATE bias reduction: PSPS often achieves zero bias even though it has a lower $R^2$ than other methods; vice-versa, even though RandomForestDRIPW and Dragonnet have the best predictive performance they fail to obtain unbiased ATE estimates.

### 4.4. Model Complexity and Parameter Tuning

PSPS builds on a $J$-state PRESS model to approximate $P(T \mid \mathbf{X})$, while simultaneously training outcome models in all $J$ states at once. The hyperparameter $J$ controls the smoothness of the learned prediction functions. The degenerate $J = 1$ case corresponds to a single outcome model with all observations falling in the same global state. As $J$ increases, the propensity score distribution can be approximated more closely while the sample size per state decreases at a rate of $J^{-1}$. Practitioners thus face a trade-off between better approximating $P(T \mid \mathbf{X})$ by increasing $J$ versus reducing $J$ to avoid over-parameterization of the full model – especially the $J$ outcome models ($\theta_1, \ldots, \theta_J$).

To explore how the choice of $J$ affects PSPS performance we undertake a simulation study where we train PSPS models with $J = 1$ (degenerate), 2, 3, 5, 10, and 20. Figure 5 illustrates that as long as $J$ is in a *reasonable* range of $O(1) - O(10)$ the causal estimation performance varies only slightly.[7] This aligns with the theoretical underpinning of PRESS as a kernel smoother that is quite robust to large $J$: as long as $J$ is sufficiently large to obtain good coverage on the unit (propensity score) interval, the actual choice of $J$ is secondary to obtain well-performing causal estimates.

---

7. See Appendix B.4 for detailed results.

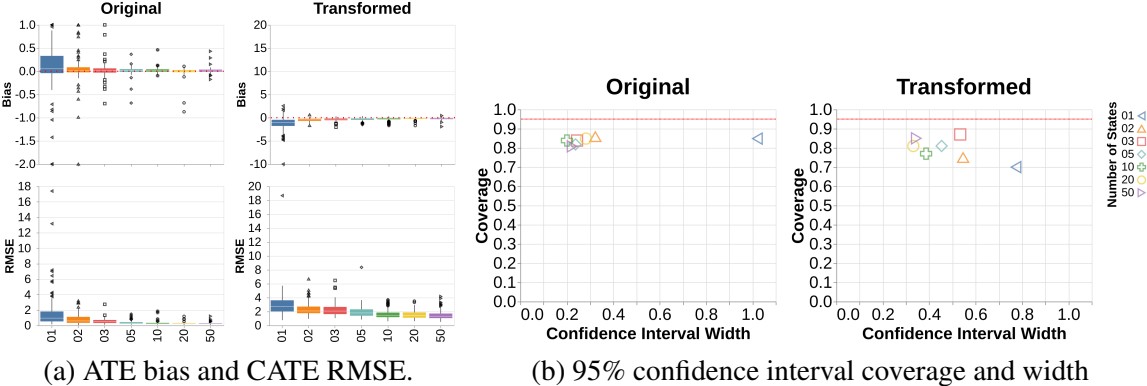

(a) ATE bias and CATE RMSE.

(b) 95% confidence interval coverage and width

Figure 5: Model complexity vs. performance for Kang & Schafer (Section 4.1).

Simulations further showed that a more complex propensity model (e.g., several hidden layers feeding into predictive states) combined with simple (e.g., linear) outcome models is sufficient to get unbiased causal effect estimates. We also found that shallow networks with standard activation functions help reduce variance of treatment effect estimates. While it remains a task for future work to explore whether PSPS is in fact a double-robust estimator (Robins and Rotnitzky, 1995), we hypothesize that these findings can be explained by double-robustness as it is clearly preferable to get *one* propensity model right, rather than trying to specify *all J* outcome models correctly.

These results suggest a practical guideline on how to use PSPS: if ATE estimates do not change for $J = 1$ vs. $J > 1$, a simple (non-linear) outcome model might be sufficient. This can be seen when comparing PSPS with $J = 1$ to other (non-PSPS) outcome model alternatives in each scenario. Interestingly, the coverage probability stays roughly constant for varying $J$, yet the confidence interval width decreases significantly as $J$ increases. We hypothesize this is a result of lower variance estimates through the better outcome model approximation. If practitioners are interested in obtaining narrow confidence intervals, we recommend increasing the number of states. Our findings are based entirely on the scenarios simulated here. For real world datasets we recommend to run cross-validation and choose that $J$ which maximizes the out-of-sample likelihood in (13).

## 5. Limitations

We note the lack of asymptotic theoretical guarantees such as consistency, whether PSPS is doubly robust or if it achieves the asymptotic minimum variance. The focus of this work is to provide a novel, powerful causal inference algorithm and evaluate how it performs on standard causal datasets. Since achieving asymptotic minimum variance does not necessarily reflect performance in practice, establishing empirical qualities of an algorithm is of interest and beneficial in its own right.

If a method is not doubly robust or consistent it may not be a viable alternative and we recognize the need for such theoretical guarantees. We hypothesize, however, that PSPS may indeed be consistent under an ignorable treatment assignment Rosenbaum and Rubin (1983b). The form of (15) is comparable to the stratification estimator using within-stratum regression in Lunceford and Davidian (2004), where consistency was proved under correctly specified propensity and regression models. Due to PSPS using neural networks with at least one hidden layer for both the propensity and regression models it can, with a reasonably sized number of hidden states, leverage the

Universal Approximation Theorem (Hornik et al., 1989), which states that neural networks can approximate any Borel measurable function. Hence, under reasonable assumptions PSPS should have correctly specified propensity and regression models, which – leveraging Lunceford and Davidian (2004) – should be sufficient to establish doubly robustness. Moreover, the form of the estimator in (15) then suggests a consistency argument. Formal proofs are left for future work.

## 6. Discussion

We introduce *predictive state propensity subclassification* (PSPS), a novel causal inference algorithm that follows the Rubin Causal Model (RCM) framework by adjusting observational causal inference through propensity models following a subclassification approach. Rather than using ad-hoc procedures to stratify units by propensity score, PSPS estimates outcome models, propensity scores, and optimal strata *simultaneously* in a joint probabilistic model. This yields efficient training algorithms that can be easily implemented in deep learning frameworks like TensorFlow. Several benchmark simulations show that PSPS has excellent statistical properties, with practically zero bias & low variance and the only method with proper confidence interval coverage compared to several state-of-the-art causal inference algorithms in the statistical and machine learning literature.

## Acknowledgments

We want to thank Penny Chu, Tony Fagan, and Jim Koehler for their ongoing support during this project. We are especially grateful to Mike Hankin, Tianqi Liu, Susanna Makela, Xiaojing Wang, and Mike Wurm for fruitful discussions and their contributions to run simulations and evaluations.

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

## Appendix A. Methodology

### A.1. Deterministic Assignment to States

Without restricting $w_{i,j}(\mathbf{X})$ Algorithm 1 is not *identical* to traditional subclassification, since each observation probabilistically maps to all states at once, rather than a deterministic $\mathbf{x}_i \mapsto s_j$ mapping. In that sense PSPS is a soft-clustering generalization of classic subclassification – akin to Gaussian mixture modeling (GMM) being a generalization of K-means (Lücke and Forster, 2019).

If practitioners want to obtain such a deterministic (hard-clustering) strata assignment, we suggest to add an entropy based penalty regularization of the form

$$\lambda_e \cdot \sum_{i=1}^{N} \left( -\sum_{j=1}^{J} w_{i,j} \log_2 w_{i,j} \right) = \lambda_e \cdot \sum_{i=1}^{N} \text{entropy}_i. \tag{18}$$

Adding (18) to (13) allows to control with how much certainty observations are mapped to predictive states. For sufficiently large $\lambda_e \gg 0$ (18) is minimized for deterministic mappings, $w_{i,j} = 0$ for all but one $j$ for each $i$, i.e., it assigns observations to exactly one state with probability 1.

We want to emphasize though that without any prior knowledge of a deterministic mapping we suggest to keep the soft-thresholding property of PRESS, as from a likelihood point of view it leads to better models. Similarly to how a GMM usually gives higher quality results compared to a hard-thresholding K-means. In practice a small $\lambda_e$ can be beneficial to make $\mathbf{w}_j$ more sparse.

### A.2. Trimming

Using predictive states in the propensity model allows us to easily assess "treatment vs. control" balance within each predictive state and then drop states from inference for which balance is deemed inadequate. A simple suggestion is to ensure that there are enough treated and control units in each state with respect to the complexity of the outcome model; alternatively balance of individual features or their products may be assessed via methods such as t-tests, etc.

## Appendix B. Simulation Studies

### B.1. Feature generation for Lunceford & Davidian

The joint distribution of $(\mathbf{X}_i, \mathbf{Z}_i)$ is given by $\mathbf{X}_{i3} \sim \text{Bernoulli}(0.2)$, then generate $\mathbf{Z}_{i3}$ as Bernoulli with $P(\mathbf{Z}_{i3} = 1 \mid \mathbf{X}_{i3}) = 0.75 \cdot \mathbf{X}_{i3} + 0.25 \cdot (1 - \mathbf{X}_{i3})$. Conditional on $\mathbf{X}_{i3}$, $(\mathbf{X}_{i1}, \mathbf{Z}_{i1}, \mathbf{X}_{i2}, \mathbf{Z}_{i2})$ is then generated as multivariate normal $N(a_{\mathbf{X}_{i3}}, B_{\mathbf{X}_{i3}})$, where $a_1 = (1, 1, -1, -1)^T, a_0 = (-1, -1, 1, 1)^T$ and

$$B_0 = B_1 = \begin{pmatrix} 1 & 0.5 & -0.5 & -0.5 \\ 0.5 & 1 & -0.5 & -0.5 \\ -0.5 & -0.5 & 1 & 0.5 \\ -0.5 & -0.5 & 0.5 & 1 \end{pmatrix}. \tag{19}$$

### B.2. Additional Simulation Study: Radcliffe & Surrey

This data generating process is taken from Radcliffe (2007) and Radcliffe and Surry (2011). Here treatment is randomly assigned to each unit with probability $p$. For each unit, $x_1$ and $x_2$ are drawn

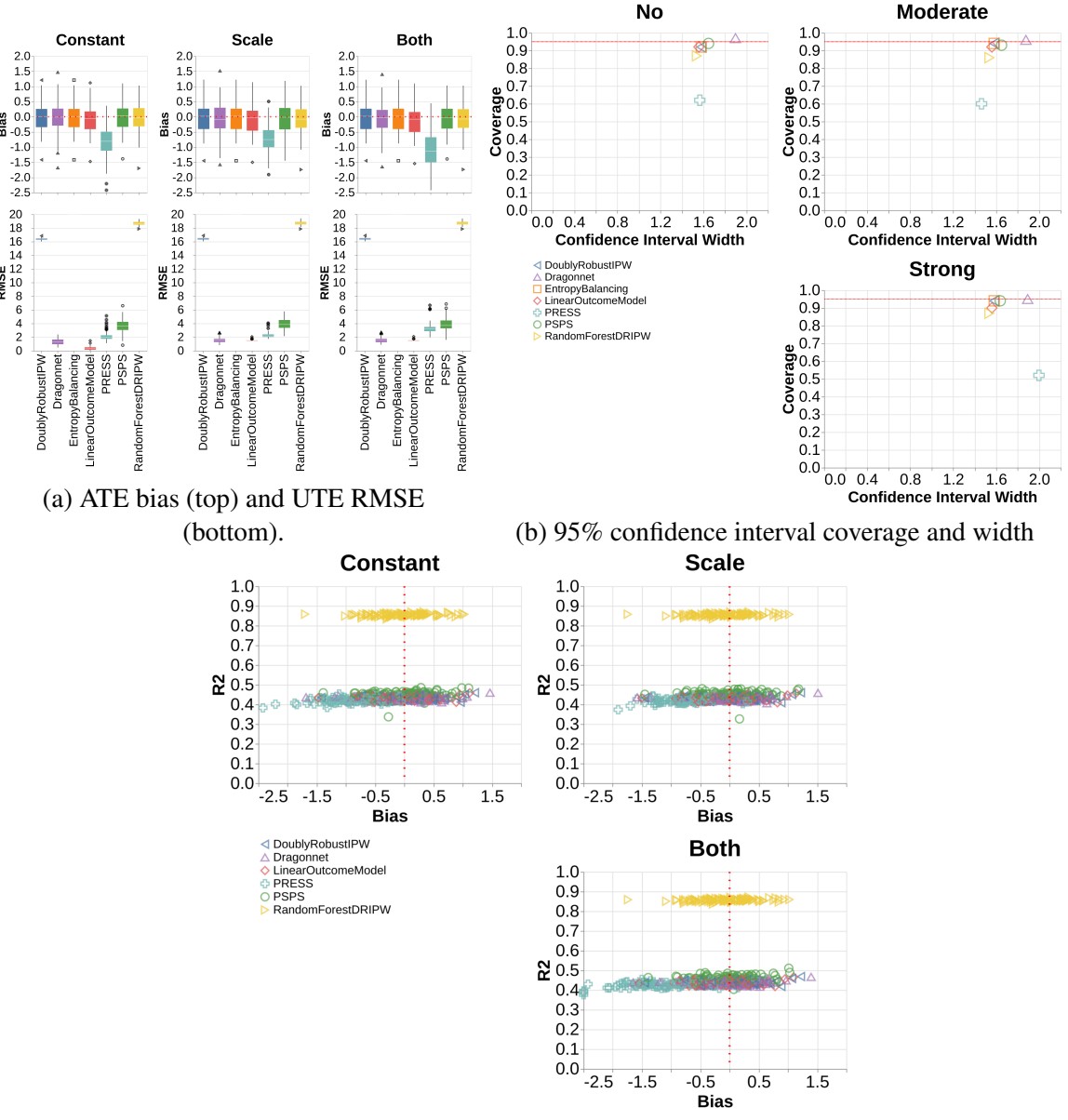

(a) ATE bias (top) and UTE RMSE (bottom).

(b) 95% confidence interval coverage and width

(c) Radcliffe & Surrey. For marginal bias distribution see sub-figure (a).

Figure 6: Model comparison for Radcliffe & Surrey dataset (Section B.2)

as independent uniform random integers in $U[0, B]$ for an upper bound $B$ (here: $B = 99$). Unobserved variables $u_1$ and $u_2$ are then drawn uniformly from $U(0, x_i)$, $i = 1, 2$ respectively. Unit-level treatment effects and outcome regressions are then set as

$$ute_i = u_{2,i} \cdot s + c \tag{20}$$

$$y_i = u_{1,i} + T_i \cdot ute_i, \tag{21}$$

where $T_i$ is the realization of a Bernoulli draw with probability $p$ (here: $p = 0.5$). This specification yields a true ATE of

$$ATE = s \cdot \mathbb{E}(u_2) + c, \text{ where } \mathbb{E}(u_2) = B/4. \tag{22}$$

In simulations we explore whether the scaling factor $s$ or shift $c$ affect the ability to recover the true treatment effect. As for Lunceford & Davidian we consider three scenarios: $const = \{s = 0, c = 3\}$, $scale = \{s = 0.1, c = 0\}$, and $both = \{s = 0.1, c = 3\}$.

Figure 6 shows that – except for the outcome PRESS neural net – all models give unbiased ATE estimates; and again, linear and Random Forest DRIPW have difficult estimating unit-level effects. Results do not vary much between "const", "scale", or "both" except for RandomForestDRIPW, which performs worse as the association grows stronger.

### B.3. Propensity Metrics and Near Positivity Violations

A common concern with propensity score methods, in particular inverse propensity weighting (IPW), are near positivity violations which are situations where the propensity to treat is close to 0 or 1 (Robins et al., 2007). Practically this means it is unlikely to find any matching control (treated) units. Statistically IPW algorithms run into near zero-division issues which result in high instability in the causal effect estimate.

We explore how estimators perform across two metrics on the propensity score distributions that measure important properties related to near-positivity violations:

- **Extremeness**: proportion of scores outside of $[0.05, 0.95]$. This highlights scenarios where the propensity model is close to near-positivity violations (Robins et al., 2007).

- **Flatness**: normalized Shannon entropy of the propensity score distribution (unconditional) (Cover and Thomas, 2006). This measures how flat (or peaked) the propensity distribution is. A score of 1.0 is a uniform distribution; a lower score corresponds to a more "peaky" distribution. Exactly zero score flatness occurs when the propensity scores reduce to singletons along the unit interval, and thus the continuous distribution degenerates to delta functions. This can happen for PSPS if the mapping from features to states ($p(s_j \mid \mathbf{X}_i)$) become deterministic, and only $J$ propensity scores in the final propensity model remain.

Across all simulations we only found that one scenario resulted in non-trivial relationships between the propensity score distribution metrics above and the quality of the ATE estimates. For the original KangSchafer dataset, without transformations, we see that DoublyRobustIPW shows various degrees of flatness and extremeness, which result in improved ATE estimates (as flatness and extremeness increased). For PSPS on the other hand all scores are away from near-one violations (extremeness $= 0.0$ for all simulations) and propensity score flatness is close to 0.0 – which corresponds to predictive state mappings that are deterministic functions from features, $\mathbf{X}$, to one of the $J$ states, $s_1, \ldots, s_J$. In either case, it had no impact on quality of ATE estimate in terms of bias.

An explanation for this lies in how PSPS uses (estimated) propensity scores to arrive at the final causal estimate: it computes a convex-combination, weighted average of the J individual outcomes models. The key here is that propensity scores are not inverted, as in any inverse-propensity weighting method, thus near-one (or near zero) propensity scores pose no problem for PSPS.

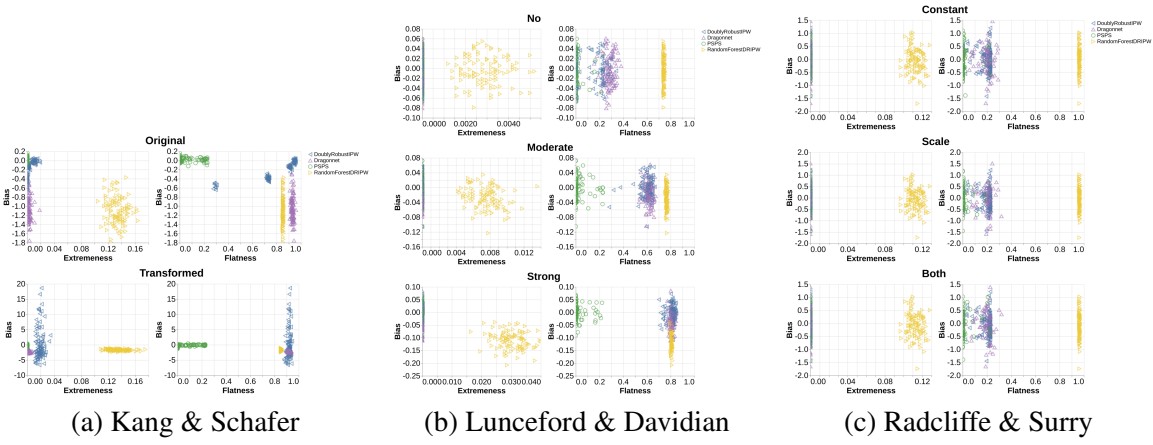

(a) Kang & Schafer     (b) Lunceford & Davidian     (c) Radcliffe & Surry

Figure 7: Propensity metrics

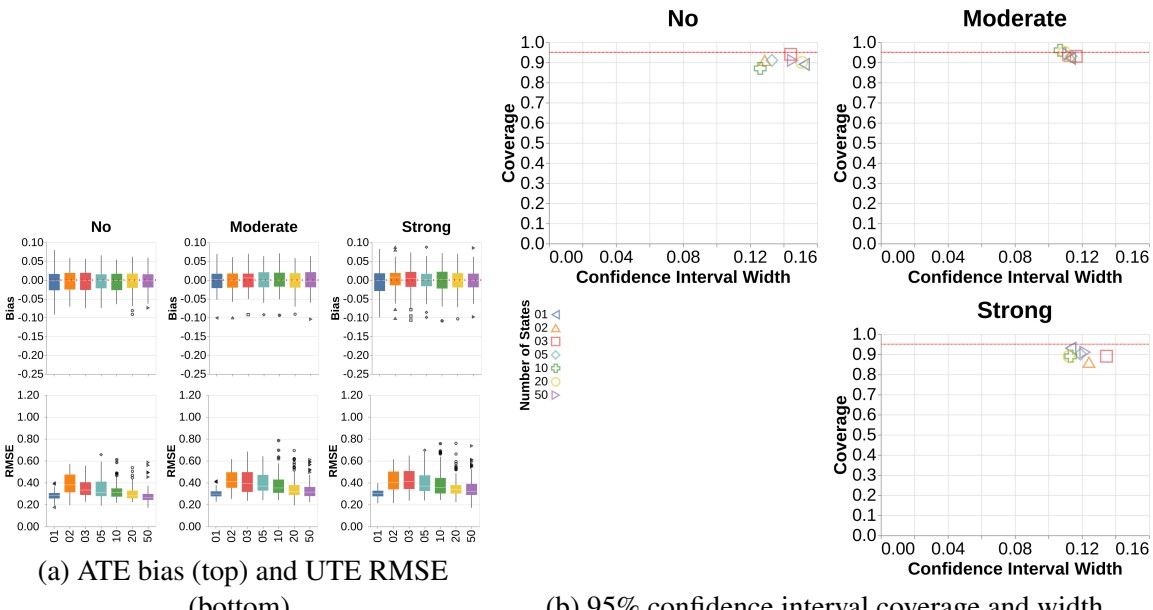

(a) ATE bias (top) and UTE RMSE (bottom).     (b) 95% confidence interval coverage and width

Figure 8: Model comparison for Lunceford & Davidian dataset (Section 4.2)

## B.4. Sensitivity Analysis For Number of States

In most scenarios $J = 10$ was sufficient, and any further increases in $J$ only yielded marginal (if any) improvements. In fact, Figure 5 shows that increasing $J$ decreases RMSE while in Fig. 8 decreasing $J$ is beneficial with $J = 10$ a natural midpoint between the two. Even a single state PSPS architecture $(J = 1)$ is competitive in certain scenarios. This is an interesting edge case as it is equivalent to a simple outcome model with averaging weights $w_{i,j} = w_{i,1} \equiv 1$ for each $i$.

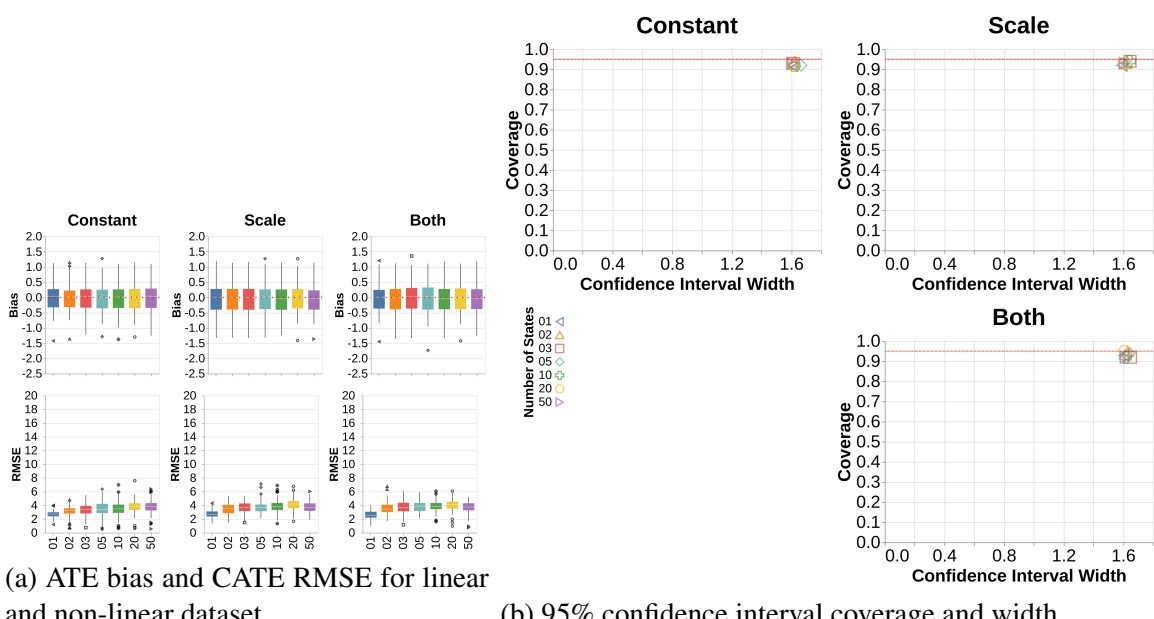

(a) ATE bias and CATE RMSE for linear and non-linear dataset.

(b) 95% confidence interval coverage and width

Figure 9: Model comparison for Radcliffe & Surry dataset (Section B.2)

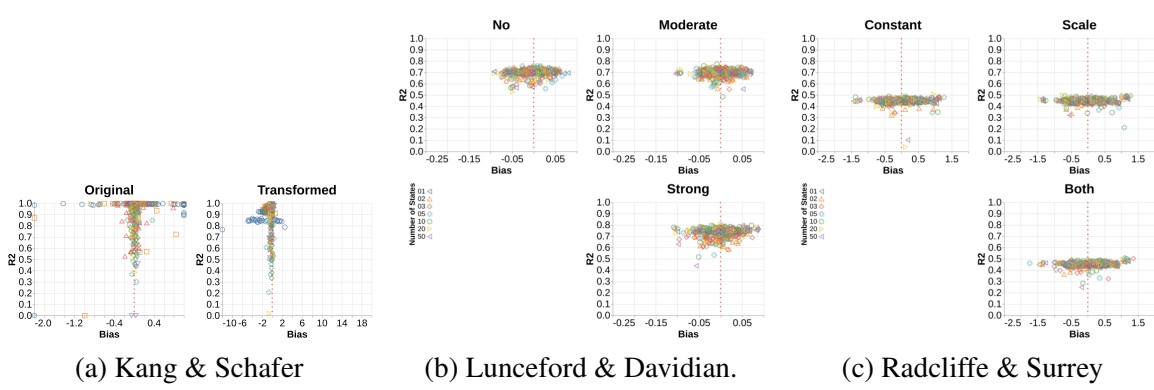

(a) Kang & Schafer

(b) Lunceford & Davidian.

(c) Radcliffe & Surrey

Figure 10: R-squared of the Outcome Model vs. ATE Bias. For marginal bias distribution see Figures 5, 8, and 6, respectively.

