# OpenReview forum: "Predictive State Propensity Subclassification (PSPS): A causal inference algorithm for data-driven propensity score stratification"
_cclear.cc/CLeaR/2022/Conference — CLeaR 2022 Poster_

### Official Review · Reviewer_9ooN · 2021-11-17

**Confidence:** 4
**Overall Score:** 7

**Main Review:**

Originality: The proposed method has two main contributions: 1) automatic selection of strata for propensity score subclassification and 2) joint optimization of the treatment and outcome model.

Significance: The proposed method addresses an important problem.

Technical quality: In the simulations, the Authors compare the proposed methodology with several other techniques. However, to see how PSPS improves estimates, the method should be compared with ad-hoc bin stratifications. This should help understand where the improvement is coming from. From PRESS on the 1st stage or from multi-task learning? What if instead of doing propensity score subclassification, one just does multi-task learning on a doubly robust estimator?
It is not clear what happens if the outcomes are not gaussian. Both simulations are based on such a data-generating process.
Are the Authors suggesting bootstrap to obtain confidence intervals also in applied settings? What is the computational burden to run the proposed method on hundreds of bootstrap samples?

Clarity: The paper reads very well.


**Summary:**

The Authors present a novel causal inference estimator based on  Predictive State Smoothing and multi-task learning.

---

### Official Review · Reviewer_2Guk · 2021-11-22

**Confidence:** 5
**Overall Score:** 4

**Main Review:**

Propensity score balancing and subclassification is not new idea in the literature. It was first proposed to deal with binary treatment effect (see e.g. Rosenbaum and Rubin, 1983) and further extended to the continuous treatment effect setting (Imai and Dyk, 2004). The authors claim that the guidance to creating the subclasses is sparse in the literature, and the existing methods are ad-hoc heuristic. The paper's main contribution is to use the Predictive State Smoothing (PRESS) (Goerg, 2017, 2018) method as a data-driven subclassification method and combine the outcome model to improve the robustness of the estimation.

The following are some comments I have on the paper:

1. In treatment effect analysis, inference on the existence of a significant effect is particularly important. Unfortunately, the paper lacks the asymptotic properties of their proposed estimator and the inference method, which weakens the value of the paper.
2. Treatment effect analysis, including the propensity score subclassification method, has been well studied and widely used in the literature for years. Therefore, more comprehensive comparisons should be conducted to show the proposed method's motivation and advantage.

    (a) For example, many works in the literature subclassify the data into five classes according to the quantiles of the propensity scores and show that this reduces the bias; see e.g. Rosenbaum and Rubin (1984); Imai and Dyk (2004); Yang et al. (2016). In addition, some established data-driven methods to determine the optimal subclasses (see e.g. Myers and Louis~2012; Orihara and Hamada, 2021). The authors should review all these methods and compare their proposed PRESS scheme to these in the simulation study.

    (b) Propensity score matching is closely related to subclassification. There, a variety of data-driven methods of measuring the distance between covariates using the propensity score are available. The authors should also compare their proposed method to these.

    (c) The PRESS method used in the proposed estimation seems to be nonparametric. If this understanding is correct, it is not surprising that it outperforms the parametric and linear models such as the linear outcome model. The authors should compare more to the nonparametric methods such as the calibration method proposed by Chan et al. (2016). Otherwise, please clarify.

3. The authors should be clear on the assumptions needed for using the propensity score balancing. The only assumption the authors mention in the paper is the ''ignorability'' in Rosembaum and Rubin (1983). However, the proof of propensity score balancing in Rosenbaum and Rubin~(1983) based on the ''ignorability'' assumption is not directly extendable to the continuous treatment case. Imai and Dyk (2004) further make the ''uniquely parametrized propensity function'' assumption to generalise the proof to the continuous treatment effect.
4. Please specify the sample size of each of the simulation studies.
5. Please enlarge all the figures in the simulation. They are too small to be readable.

**Reference**

Chan, K. C. G., Yam, S. C. P. and Zhang, Z. (2016) Globally efficient non-parametric inference of average treatment effects by empirical balancing calibration weighting. *Journal of the Royal Statistical Society: Series B*, **78**, 679 - 700.

Goerg, M. (2017) Predictive State Smoothing (PRESS): Scalable non-parametric regression for high-dimensional data with variable selection. *Technical report, Google, URL https://ai.google/research/pubs/pub46141*.

-(2018) Classification using Predictive State Smoothing (PRESS): A scalable kernel classifier for high-dimensional features with variable selection. *Technical report, Google, 2018. URL https://ai.google/research/pubs/pub46767*.

Imai, K. and Dyk, D. A. (2004) Causal inference with general treatment regimes. *Journal of the American Statistical Association*, **99**, 854-866.

Myers J.A. and Louis T.A. (2012) Comparing treatments via the propensity score: stratification or modeling? *Health Services and Outcomes Research Methodology*, **12**, 29-43.

Orihara, S. and Hamada, E. (2021) Determination of the optimal number of strata for propensity score subclassification. *Statistics \& Probability Letters*, **168**, 108951.

Rosenbaum, P. R. and Rubin, D. B. (1983) The central role of the propensity score in observational studies for causal effects. *Biometrika*, **70**, 41-55.

Yang, S., Imbens, G. W., Cui, Z., Faries, D. E., and Kadziola, Z. (2016) Propensity score matching and subclassification in observational studies with multi‐level treatments. *Biometrics*, **72**, 1055-1065.


**Summary:**

The paper introduces Predictive State Propensity Subclassification algorithm for estimating the average treatment effects and unit-level treatment effects, based on the propensity score subclassification idea in the literature.

---

### Official Review · Reviewer_Dgts · 2021-11-27

**Confidence:** 4
**Overall Score:** 4

**Main Review:**

Pros:
1.	The proposed estimators perform very well compared to the other estimators used in the simulation studies, especially the Kang & Schafer data generating mechanism.

Cons:
1.	Estimation of the average treatment effect and (to a lesser extent) conditional average treatment effect is an extremely well-studied topic with plenty of theoretical guarantees established in the literature. Thus, this is a crowded research area with a higher bar for new methods. This article provides no theoretical guarantees for its estimators – only some largely informal motivation for the form of the estimator.
2.	Given that the doubly robust augmented IPW readily admits deep learning (or any other kind of supervised learning) for the outcome regression and propensity score, this would seem to be the more fair comparison for the simulation studies. In other words, does the proposed method do well because of the particular network architecture, or is it just the deep learning that’s performing well in these scenarios?
3.	Is sample splitting/cross-fitting being used for the DRIPW? When using random forest, it is known to perform poorly without sample splitting, so this should really be used for a fair comparison.
4.	Was the bootstrap used for all estimators? The bootstrap is not theoretically justified for especially data-adaptive estimators such as those involving the bootstrap.
5.	Robins and Ritov (1997) showed that the propensity score is ancillary for the average treatment effect, and so the maximum likelihood estimator does not depend on the propensity score. Therefore, the proposed estimator is not an MLE as claimed in Section 2.5.
6.	Double robustness is not a general property of any estimator of any causal effect. One must show that their proposed estimator is doubly robust, and if there are multiple outcome models, one must show that this double robustness entails correctly specifying the propensity score or all outcome models.
7.	Overall, I did not find the writing to be very strong. There are a fair amount of vague and redundant statements.
8.	I did not understand the description of the predictive states very well. In particular, I didn’t understand how the $s$’s went from being seemingly deterministic functions of $X$ (“$\{s_1(X),\ldots,s_J(X)\}$”) to being random variables conditional on X (“$p(S=s_j\mid X)$”). Also, isn’t $\rho_{I,j}=1$ by assumption (“$p(T\mid X_{i_2}=p(T\mid s_j)$”)?
9.	“Natural experiment” is used where I believe “observational study” would be more appropriate.
10.	I understand that predictive performance does not necessarily translate to good estimate of the average treatment effect (this is known from literature on semiparametric inference and undersmoothing), but I am a bit surprised by the level of disconnect between $R^2$ and UTE estimation performance.

Robins, J. M., & Ritov, Y. A. (1997). Toward a curse of dimensionality appropriate (CODA) asymptotic theory for semi‐parametric models. Statistics in Medicine, 16(3), 285-319.


**Summary:**

This article proposes new estimators of the average treatment effect and unit-level/conditional average treatment effect using neural networks. Simulation studies demonstrate favorable performance against some existing estimators.

---

### Official Review · Reviewer_tkA2 · 2021-11-30

**Confidence:** 3
**Overall Score:** 6

**Main Review:**

_Originality._ The method provides a novel approach to data-driven propensity subclassification.

_Significance._ PSPS addresses an important problem in an interesting way by integrating the propensity and the causal effect estimation in a probabilistic model using PRESS. The method is shown to work better than existing methods in some simulation scenarios.
In my opinion, the lack of a sample implementation is a limitation to the applicability and use of this method by practitioners. The probabilistic model and loss functions are simple and clearly described, but the PRESS model on which this method relies is not well known. The paper would benefit from including an open source implementation of the code used to run the experiments both to facilitate use in other contexts and reproducibility.

_Technical quality._ In general, the paper is technically sound. The description of the method is clear and correct, the evaluation metrics are adequate and the provided sensitivity analyses address all major considerations. I have however highlighted some points that could be improved:

- The guidance for the choice of the number of states parameter (_J_) in the main text could be improved by restructuring it to include the most striking observations from appendix B.2. I suggest combining sections 2.8 and 4.4 and referring to the sensitivity analysis from B.2. which I found insightful.
- Footnote 5 describes the choice of parameters for Dragonnet which seem to only be geared at making it comparable to PSPS in terms of the number of parameters. How does the model perform when using the hyperparameter/architecture settings recommended by the authors? A more transparent report could include this as a comparison and nuance the effect of the number of parameters as appropriate in the discussion.
- A comparison with _ad hoc_ but better known methods for propensity subclassification would be an asset.
- I believe there is an error in the summation indices in equation (3)

Clarity. Overall, the submission is clear and well written. In terms of wording, I prefer to limit the use of the term “natural experiments” to specific instances of observational studies where there is a notion of experiment or intervention (_e.g._ instrumental variable, regression discontinuity or similar designs). This definition is also used by others (see Dorie V, _et al._ Statistical Science 2019).

In their current state the figures are hard to read and so legends could be more self explanatory.

Some links seem to be broken:
- Reference to Figure B in Figure 5 legend



**Summary:**

The paper introduces a new algorithm called PSPS to estimate causal effects using an approach akin to propensity subclassification.

---

### Author Response · Authors · 2021-12-03
**General response**

We would like to thank the reviewers for their constructive feedback. Below we have responded to each reviewer with the changes we will make to the paper to address their comments.

Minor changes:
- We will change natural experiments to observational studies.
- We will increase font sizes for titles & labels in figures.
- Thank you for pointing out minor typos or broken links; we will fix those.

We will respond to each review separately, except for a common theme that all reviewers had we address here: why we chose to not compare PSPS to a version of a traditional sub-classification method.

We appreciate the suggestions to compare to a wider range of models with perhaps a greater focus on more traditional versions of subclassification. The current set of comparisons were chosen as we found they represented a good selection of both ML and non-ML type popular methods whilst being able to be readily embedded into our simulation framework that can be run at scale. A good suggestion from 2Guk, for example, was to compare a method that subclassifies on the quantiles of the propensity score into 5 groups as per Rosenbaum and Rubin (1984); Imai and Dyk (2004); Yang et al. (2016). This method, however, also requires estimation of the propensity score and hence any comparison would also depend on the type of propensity score model chosen. A popular method such as iteratively fitting logistic regressions and manually adding/removing covariates to assess balance is not feasible for our simulation.

While we agree that theoretical properties of asymptotic of PSPS are interesting to investigate, we chose to focus on empirical comparison of PSPS compared to other algorithms, particularly as algorithms with good asymptotic properties can still fail for ML practitioners in empirical settings.  We leave asymptotic theory for future work.

---

### Decision · Program_Chairs · 2022-01-12

**Decision:**

Accept (Poster)

**Comment:**

The paper proposes a causal inference algorithm for automatic selection of strata for propensity score subclassification and joint optimization of the treatment and the outcome model. Empirically, the proposed method performs better as compared to other baselines in a small scale simulation study. The paper got mixed reviews because of the absence of theoretical guarantees. The authors also release an implementation of the proposed method. I encourage the authors to include a "limitations" section discussing the limitations of the proposed method (and also the absence of theoretical guarantees). It would be important highlight the empirical contribution as well as absence of theoretical guarantees.